



# Mobile open dynamic chamber measurement of methane macroseeps in lakes

Frederic Thalasso[1], Katey Walter Anthony[2,3], Olya Irzak[4], Ethan Chaleff[4], Laughlin Barker[4], Peter Anthony[2], Philip Hanke[2], Rodrigo Gonzalez-Valencia[1]

[1]Biotechnology and Bioengineering Department, Cinvestav, Avenida IPN 2508, Mexico City, 07360, Mexico
[2]Water and Environmental Research Center, University of Alaska Fairbanks, Fairbanks, Alaska 99775, USA
[3]International Arctic Research Center, University of Alaska Fairbanks, Fairbanks, Alaska 99775, USA
[4] Frost Methane, Oakland, California 94612, USA

*Correspondence to*: Frederic Thalasso (thalasso@cinvestav.mx); Katey Walter Anthony (kmwalteranthony@alaska.edu)

**Abstract.** Methane ($CH_4$) seepage; i.e., steady or episodic flow of gaseous hydrocarbons from subsurface reservoirs, has been identified as a significant source of atmospheric $CH_4$. However, radiocarbon data from polar ice cores recently brought into question the magnitude of fossil $CH_4$ seepage naturally occurring. In northern high latitudes, seepage of subsurface $CH_4$ is impeded by permafrost and glaciers, which are under an increasing risk of thawing and melting in a globally warming world, implying the potential release of large stores of $CH_4$ in the future. Resolution of these important questions requires a better constraint and monitoring of actual emissions from seepage areas. The measurement of these seeps is challenging, particularly in aquatic environments, because they involve large and irregular gas flowrates, unevenly distributed both spatially and temporally. Large macroseeps are particularly difficult to measure due to a lack of lightweight, inexpensive methods that can be deployed in remote Arctic environments. Here, we report the use of a mobile chamber for measuring emissions at the surface of ice-free lakes subject to intense $CH_4$ macroseepage. Tested in a remote Alaskan lake, the method was validated for the measurement of fossil $CH_4$ emissions of up to $1.08 \times 10^4$ g $CH_4$ m$^{-2}$ d$^{-1}$ (13.0 L m$^{-2}$ min$^{-1}$ of 83.4 % $CH_4$ bubbles), which is within the range of global fossil methane seepage and several orders of magnitude above standard ecological emissions from lakes. In addition, this method allows for low diffusive flux measurements. Thus, the mobile chamber approach presented here covers the entire magnitude range of $CH_4$ emissions currently identified, from those standardly observed in lakes to intense macroseeps, with a single apparatus of moderate cost.

## 1 Introduction

Methane ($CH_4$) is a powerful greenhouse gas, that contributes about 20 % of the warming induced by greenhouse gases (Kirschke et al., 2013), with a global emission estimated to 572 or 737 Tg $CH_4$ yr$^{-1}$, for top-down or bottom-up budget estimations, respectively (Saunois et al., 2020). In addition to biotic and industrial sources, gas seepage; i.e., steady or episodic flow of gaseous hydrocarbons from subsurface sources to the surface, has been identified as a significant source of





atmospheric $CH_4$, estimated to range 42−76 Tg $CH_4$ yr$^{-1}$ (Schwietzke et al., 2016; Etiope et al., 2019). However, recent analysis of atmospheric $CH_4$ radiocarbon ages in polar ice cores suggests that fossil $CH_4$ seepage of natural origin is at least an order of magnitude lower (Hmiel et al., 2020). Thus, $CH_4$ seepage, which has been classified into microseepage and macroseepage (i.e., diffuse exhalation and channeled flows, respectively), and previously thought to be a major component

of the global $CH_4$ cycle, needs to be better constrained.

In northern latitudes, large quantities of geologic $CH_4$ are trapped by permafrost and glaciers, which form a "cryosphere cap" that restricts their flow to the atmosphere. Given that the Arctic is exposed to greater climatic warming than other latitudes (Trenberth et al., 2007; Post et al., 2019; Ito et al., 2020), the disintegration of the cryosphere cap would lead to transient release of $CH_4$ along faults (Spulber et al., 2010), through glacier retreat (Lamarche-Gagnon et al., 2019) and/or through

permafrost thawing (Walter Anthony et al., 2012). The magnitude of that potential $CH_4$ release is unknown, but a carbon store of over 1,200 Pg trapped by the cryosphere cap has been estimated (Isaksen et al., 2001; Flores et al., 2004; Gautier et al., 2009; McGuire et al., 2009; Collett et al., 2011). Hence, the conversion and release of a small fraction of that carbon to $CH_4$ may represent a significant input to the current atmospheric $CH_4$ pool, estimated to 5 Pg (Isaksen et al., 2001; Engram et al., 2020).

It is therefore of utmost importance to locate and accurately quantify $CH_4$ seepage. In aquatic environments, the location of $CH_4$ seepage can be determined by direct gas measurements or through visual observation of ebullition; i.e., outburst of gas. In northern latitudes, seasonal ice cover provides an unique opportunity to more accurately assess $CH_4$ ebullition, since the physical impact of bubbles breaking at the water surface delays ice formation, resulting in bubble-induced open holes during early winter (Walter Anthony et al., 2012). Later during the winter, ebullition also results in heterogeneous ice cover with

gas inclusions that can be detected by remote detection methods such as satellite SAR sensing (Engram et al., 2020).

The quantification of $CH_4$ macroseepage to the atmosphere is challenging, because it involves irregular gas flowrates, which, taken individually, can range from few milliliters to tens of liters per minute (Walter Anthony et al., 2012), and are spatially and temporally unevenly distributed. When the gas flow rate is low, measurements are standardly done through quantifying the gas accumulation within a chamber, located at the surface of the ground (Rolston, 1986) or water (Etiope, 2015). This

approach, termed "closed chamber technique" in reference to the absence of a gas flowing through the chamber, is simple and easy to deploy and can be automated through a programmed venting device and a continuous gas analyzer, allowing repeated measurements over long-term periods without supervision (Davis et al., 2018; Martinsen et al., 2018). However, closed chambers have a fixed volume, in such manner that they are efficient for diffusive flux measurements combined with small ebullitive events (Tang et al., 2017), but are not applicable to large ebullitive fluxes. Indeed, the chamber capacity to

capture gas volumes is limited and any significant gas volume input, surpassing the chamber capacity, would escape from below the chamber, at the chamber/water interface.

An alternative method for the measurement of individual macroseeps is the use of submerged bubble traps. These consist of an inverted funnel, of variable design, placed underwater, in which the gas bubbles are collected during their ascent. This direct, simple and robust method is widely used for the determination of microbial $CH_4$ ebullition in aquatic ecosystems



(Walter et al., 2008; Wik et al., 2013; Delwiche and Hemond, 2017), and some automated designs have been suggested (Varadharajan et al., 2010; Walter Anthony et al., 2010; Maher et al., 2019). Among these systems, very recently, Duc et al. (2020) have suggested a new design of an automated chamber, which allows for the autonomous measurement of diffusive and ebullitive $CH_4$ and $CO_2$ fluxes, based on an inverted funnel with a pressure-based automatic counting and release of the captured bubbles. However, bubbles traps are limited in size and capacity since maneuverability and avoidance of heavy

counterweights to compensate for the trap buoyancy are a concern (Bowen et al., 2008). Thus, these systems are well adapted for standard episodic bubble releases but are probably not the best option for macroseeps with large gas flowrates.

A third method, commonly used, is an approximation of the flowrate to the atmosphere by examining the size and frequency of individual bubble trains, based on previous experiences and observed correlations (Etiope et al., 2004), or after field calibration of individual seeps with bubble traps (Walter Anthony et al., 2012). As described in the latter, this high

throughput method allowed for the quantification of thousands of individual seeps in northern lakes, through terrestrial and remote aerial observations. However, this method is probably subject to a large error, since for instance, a 10 % appreciation error in bubble diameter returns a 33 % error in bubble volume, and therefore in its $CH_4$ flux. In addition, bubble counting of fast rising trains is not an easy task.

Although unreported so far for macroseepage in lakes, a fourth potential method is the use of underwater echo-sounders,

which allow for bubble counting together with the determination of their sizes and rising speeds (Ostrovsky et al., 2008; DelSontro et al., 2011; McGinnis et al., 2011). However, to the best of our knowledge, this method has been applied to $CH_4$ seepage in marine environments (Jansson et al., 2019) but not to lake macroseepage. Despite their undeniable potential, these methods still present some uncertainty in quantifying gas emissions (Ostrovsky et al., 2008; DelSontro et al., 2015) and their applicability to intense trains of bubbles with a large size distribution is uncertain and would probably require intensive field

research before they could be validated. Moreover, hydroacoustic methods are limited to the ebullitive component of $CH_4$ emissions, thus a complementary method is required if diffusive flux needs to be quantified.

A fifth method, based on floating chamber through which a $CH_4$-free carrier gas is continuously flowing, has been recently proposed (Gerardo-Nieto et al., 2019). That method, called Open Dynamic Chamber (ODC) allows for the combined and continuous monitoring of diffusive and ebullitive flux, but is limited to conditions where the ebullitive flow rate is relatively

small, compared to the carrier gas flowrate. These conditions impede the deployment of the ODC for large ebullitive fluxes like macroseepage.

There is therefore still a need for a field deployable method for flux determination of macroseepage in aquatic ecosystems, not only to update current seepage estimations, but also to monitor their expected progression. Preferably, the method should present the following attributes: (1) low-cost, small, light and robust, for easy field deployment in remote locations, (2) with

a high throughput capacity including large spatial coverage, and (3) able to measure $CH_4$ emissions in a large span of conditions, from low diffusive flux to large stochastic ebullitive events, with a single apparatus. The objective of this work was to develop a chamber design, the Mobile Open Dynamic (MOD) chamber, which fulfills these attributes, and to test it in





a northern lake where large ebullition seeps have been detected. The accuracy of the MOD chamber was assessed through the parallel deployment of a 45 m$^2$ bubble trap.

## 2 Methods

### 2.1. MOD chamber

A rational approach to measure emissions from strong ebullitive seepage is to measure a sample of individual seeps and to multiply the measured emissions by the seep number (Walter Anthony et al., 2012). However, as we will show hereafter, precisely measuring the emission of individual seeps in aquatic environments is a difficult task. Based on field experience,
including testing several chamber designs and shapes, our trial/error approach gave birth to the Mobile Open Dynamic (MOD) chamber. The concept of the MOD chamber is based, similarly to the broadly used closed chamber, on the capture of the gas emitted from the lake into a cavity connected to a CH$_4$ detector. However, in the MOD chamber, a continuous air flow through the chamber is maintained and the CH$_4$ concentration is continuously monitored in the gas exiting the chamber. The concept of the open chamber; i.e. through which a carrier gas is flowing, was initially suggested by Edwards and Sollins
(1973) for the measurement of emission in soils and adapted later by Gerardo-Nieto et al. (2019) for aquatic ecosystems. However, three major differences distinguish the MOD chamber concept from these previous works; (i) atmospheric air is used as carrier gas, as opposed to CH$_4$-free nitrogen gas, avoiding the requirement of heavy compressed gas cylinders, (ii) the concept and mass balance of the MOD chamber allows quantification of CH$_4$ emissions with a significant volumetric flowrate, and (iii) the chamber is designed to be mobile; i.e., in motion during continuous transects measurement. We
engineered chamber mobility for several important reasons. First, we will show that positioning a measurement apparatus exactly over a macroseep is very challenging in the absence of a stable ice platform, not only because any floating device is subject to a constant and unavoidable movement caused by wind and waves, but also because the seep itself causes water convection at the lake surface that pushes the trap off of the bubbling hotspot. In motion, a well-designed chamber can cross macroseeps without being diverged. The second reason is that the measurement of macroseeps involves high ebullition
entering the chamber, while a mobile chamber passing over a macroseep accumulates a limited quantity of gas during the short period of time, and as we will show hereafter, these conditions avoids the requirement of a gas flow rate measurement, which is technically very challenging with stochastic and discontinuous ebullition. The third reason is that strong ebullition in lakes has been repeatedly reported to occur in fixed locations (Walter et al., 2006; Walter Anthony et al., 2012) (e.g. point-source seeps; illustrated by Fig. S1), each one with a specific relative overall magnitude. Thus, an approach based on
measuring individual seeps and counting, is a relatively easy task but requires an arbitrary classification, while transecting at constant speed gives the same specific weight to all measurements done along that transect, including low-flux and hotspot-flux seeps of variable magnitude. Despite these potential benefits, we acknowledge that the chamber motion has undoubtedly some effect on the gas/liquid boundary layer at the surface of the lake (Schubert et al., 2012; Lorke et al., 2015), which in its





turn may modify substantially the diffusive flux captured by the chamber, compared to a steady interface. This effect was

quantified during field testing.

The conceptual sketch of the MOD chamber is presented in Fig. 1; specific dimensions are presented in Fig. S2. This aluminum chamber consists of two half-tubes separated by a short distance, serving as a double hull, oriented perpendicular to the main longitudinal axis; i.e., motion axis. An aluminum sheet laterally covers these half-tubes as well as the top of the space between them, thus defining a chamber volume. The bottom of this headspace is open to collect gas emitted from the

lake. The use of two half tubes in contact with water avoids straight angles along the motion axis. Together with a limited draft, this design minimizes mixing of the water surface while moving at low speed. Therefore, it reduces the effects of motion on the air/water boundary layer. In addition, the lateral aluminum foils fulfill the function of a double keel, reducing lateral movement while crossing hotspots. A fixed flowrate of the chamber content is extracted to a $CH_4$ detector. A purging vent allows for pressure equilibration and an electric fan located inside the chamber ensures homogeneity of the gas phase.

## 140 2.2. Mass balance of the MOD chamber

Whatever the type of emission; i.e., diffusive or ebullitive, the $CH_4$ mass balance of the MOD chamber can be described by Eq. (1), which considers a $CH_4$ input caused by lake emissions (first term), a $CH_4$ output caused by the $CH_4$ analyzer extraction flowrate (second term), and an input or output of gas through the purge (third term), used to equilibrate pressure between the chamber and the atmosphere (See Fig. 2 for details)

$$\frac{dC_C}{dt} = F \cdot \frac{A_C}{V_C} - \frac{Q_D}{V_C} \cdot C_C + \frac{Q_i}{V_C} \cdot C_i \qquad (1)$$

where $C_C$ is the $CH_4$ concentration in the headspace of the chamber (g m$^{-3}$), which is considered completely mixed, $F$ is the $CH_4$ flux emitted by the lake, diffusive and/or ebullitive (g m$^{-2}$ s$^{-1}$), $A_C$ is the area of the chamber in contact with the water surface (m$^2$), $V_C$ is the volume of the chamber (m$^3$), $Q_D$ is the air flowrate extracted from the chamber to the detector (m$^3$ s$^{-1}$); $Q_i$ and $C_i$ are respectively the gas flowrate (m$^3$ s$^{-1}$; positive for incoming flow) and the $CH_4$ concentration (g m$^{-3}$), entering or

exiting the chamber through the purge.

The pressure equilibration guaranteed by the purge, results in a constant chamber volume and dictates that its flowrate; $Q_i$, depends on the ebullitive gas flowrate ($Q_0$) and the gas flowrate extracted by the detector ($Q_D$), which is described by Eq. (2):

$$Q_i = Q_D - Q_0 \qquad (2)$$

Thus, if the $Q_0$ captured by the chamber is greater that the $Q_D$, the resulting $Q_i$ is negative; i.e., the purge allows headspace gas to flow out of the chamber, and $C_i$ is equal to $C_C$. On the contrary, if $Q_0$ is smaller than $Q_D$, the resulting $Q_i$ is positive; i.e., the purge allows atmospheric air to flow into the chamber, and $C_i$ is equal to the atmospheric $CH_4$ concentration; $C_{ATM}$. By substituting $Q_i$ in Eq. (1), we obtain a general mass balance equation:

$$\frac{dC_C}{dt} = F \cdot \frac{A_C}{V_C} - \frac{Q_D}{V_C} \cdot C_C + \frac{Q_D - Q_0}{V_C} \cdot C_i \qquad (3)$$



Considering that the chamber would be constantly in motion, passing persistently over low-flux and hotspot-flux ebullition
seeps, the mean ebullitive flowrate captured would be reduced, and the condition $Q_D < Q_0$ may occur. Under this scenario, $Q_i$
would be positive (Eq. (2); gas flowing-in the chamber through the purge), $C_i$ is replaced by $C_{ATM}$, and Eq. (3) becomes:

$$\frac{dC_C}{dt} = F \cdot \frac{A_C}{V_C} - \frac{Q_D}{V_C} \cdot C_C + \frac{Q_D - Q_0}{V_C} \cdot C_{ATM} \qquad (4)$$

During field deployment of the chamber, we observed that $CH_4$ accumulated rapidly into the chamber, reaching a

concentration several order of magnitude above atmospheric air ($C_C > C_{ATM}$). In addition, ($Q_D$ - $Q_0$) is inevitably lower that
$Q_D$, in such manner that the third term of Eq. (4) is negligible compared to the second term. Thus, after simplification, and
solving Eq. (4) for $F$, we obtain:

$$F = \left(\frac{dC_C}{dt} + \frac{Q_D}{V_C} \cdot C_C\right) \cdot \frac{V_C}{A_C} \qquad (5)$$

It is important to note that in Eq. (5) the only flowrate required is $Q_D$, which is a design parameter; i.e., flowrate of the

detector's internal pump, which can be easily measured and calibrated. Thus, through motion of the chamber, the conditions
where $Q_0 < Q_D$ is beneficial for data interpretation as it avoids the requirement of $Q_0$ determination. The mean flux ($\bar{F}$)
emitted by the lake over a time lapse ($\Delta t$) is given by Eq. (6), where $\overline{C_C}$ is the mean $C_C$ measured, and which can be easily
numerically solved and eliminates the impact of the measurement noise.

$$\bar{F} = \left(\frac{\Delta C_C}{\Delta t} + \frac{Q_D}{V_C} \cdot \overline{C_C}\right) \cdot \frac{V_C}{A_C} \qquad (6)$$

**2.3. Chamber design**

Figure S2 shows the dimensions of the MOD chamber prototype and Fig. 3 shows the prototype while being operated. The
8.56 kg chamber was formed and welded from 3/32″ (2.3 mm) thick aluminum sheet. Its total length was 95 cm long. The
draft of the chamber was limited to 6 cm, and because of the cylindrical hull, the buoyancy (kg) was a power function of the
draft (cm; buoyancy = $0.67 \cdot Draft^{1.48}$), ensuring better vertical stability. It is worth noting that the pressure inside the chamber

was in equilibrium with the atmospheric chamber, through the 1" purge tubing (see below), thus ensuring a constant draft
and chamber volume and area. The open contact area with water was 0.109 m$^2$ and the volume of the chamber was 10.8 L.
The chamber content was homogenized with a 2″ battery operated fan (explosion proof), located inside the chamber volume.
The chamber was connected to a $CH_4$ detector; we used either an EX-TEC (HS 680, Sewerin, Germany), or an ultraportable
greenhouse gas analyzer (UGGA, Los Gatos Research, CA). Both detectors had a data acquisition frequency of 1 Hz. The

EX-TEC featured a gas sampling rate ($Q_D$; internal pump) of 1.4 L min$^{-1}$, a $CH_4$ lower detection limit (LDL) of 1 ppmv and a
measurement range of 0–100 % v/v. The UGGA, equipped with the extended range option, included a gas sampling rate of
1.1 L min$^{-1}$ ($Q_D$), and a $CH_4$ concentration measurement range of 0−10 % v/v with a $CH_4$ LDL of 0.01 ppmv.

In addition, the chamber was also equipped with a purge, which consisted of a 1″ internal diameter steel tubing, a water trap
and an airflow sensor with high sensitivity at low flow rates (AWM720P1, Honeywell, Mexico). This flow sensor's

measurement range was 0 to 200 L min$^{-1}$ for both positive and negative flows, corresponding to a pressure drop of 0 to 2.74





mbar. To reduce wind effects on the airflow sensor, the output of the sensor was covered with open-cell polyurethane foam. In order to reduce weight and thus the draft of the chamber, the MOD chamber was attached with a loose line and a stick, at the front of a small inflatable Alpacka Packraft, where the operator was, and in which the detectors and the batteries were placed.

## 2.4. Site description

The MOD chamber was tested at Esieh Lake (informal name), a 6.4 ha lake, located 40 km north of Kotzebue in north-western Alaska (67.249, -162.735) on August 21–26, 2018. At the time of the measurement campaign, Esieh Lake was characterized by sections with constant and intense ebullition (i.e. macroseepage), surrounded by large areas where no ebullition was observed. In addition, some calibration experiments were done at ''Lago de Guadalupe'' (LG; 19.633, -99.260), a 450 ha subtropical dendritic reservoir located at 2300 m above mean sea level, 25 km north of the Mexico City limits. This lake was characterized by a relatively small western section with moderate ebullition and an eastern section, in which no ebullition was observed.

## 2.5. Flux measurements and data interpretation

A section of Esieh Lake with intense ebullition was used to test the MOD chamber. The procedure was as follows; (1) the chamber was lifted out of the water and ventilated for one minute, (2) then the chamber was gently positioned on the surface of the water, (3) the chamber was immediately put in motion, at a speed of approximately 0.30 m s$^{-1}$, while (4) the detector was continuously recording CH$_4$ concentration, and a GPS (GPSmap 76 CSX, Garmin, USA) was recording the monitoring track. Measurement and motion were maintained, until a significant CH$_4$ concentration increase was observed in the chamber, usually reaching the volume percent range, or when the zone with high ebullition was traversed. This procedure was repeated for a total of 15 transects during the field campaign. In the section of the lake where no ebullition was observed, the MOD chamber was also tested for diffusive flux measurements. With that purpose, the same strategy was used, except that the chamber was maintained stationary, the purge was closed and the CH$_4$ detector was connected in a loop. CH$_4$ measurements were started about 30 s after positioning the chamber on the lake surface, to allow for equilibration, and measurement was sustained for 3 minutes each. During the field campaign, a total of eight triplicate diffusive flux measurements were done.

As it is shown in the Results and Discussion section, during transects or stationary measurements, a continuous increase of the CH$_4$ concentration read by the detector ($C_D$) was observed, as expected, with some abrupt rises when bubbles entered the chamber. The data interpretation (described in detail in section S1) included the conversion of $C_D$ data from ppmv (as read by the detectors) to mass units (g m$^{-3}$), using the ideal gas law. Then, the determination of CH$_4$ concentration in the chamber ($C_C$) was determined, from $C_D$, taking into account the response time of the system, determined in the field. Indeed, even if it can be assumed that a bubble entering the chamber is immediately mixed within the chamber, the detectors have an inherent response time ($\theta$). This effect causes a certain delay and a buffer time, between the actual concentration read by the detector





$C_D$ and $C_C$. To take this delay into account a standard mixing model was used during the data treatment process, which is described in detail in the supporting information section (Section S1). From $C_C$ data, flux time series were established, after

data smoothening (Section S1). In addition to $F$ determinations, the step increases of $C_C$ were exploited to determine the $CH_4$ content of the bubbles ($M_B$), according to a simple procedure, as well as the volume of the bubbles ($V_B$) and their equivalent spherical diameter ($d_B$), according to eqs S5 and S6, after determining the $CH_4$ bubble percentage of 83.4 % v/v (see results section).

For mapping purposes, to avoid interpolating large dataset (one $F$ and one location for each second of measurement), each

transect was segmented into 3 to 5 subdivisions, typically 10 m long. The mean flux for each segment was determined according to Eq. (6), and the central coordinates of the segment was considered for mapping. A map of $CH_4$ emissions was established from $F$ data interpolation using Surfer 11.0 software (Golden Software, USA). The selection of the best interpolation method among Kriging, Minimum Curvature, Inverse Distance to a Power, Radial Basis Function, and Local Polynomial was based on two criteria: the mean absolute error and the mean bias error (Wilmott and Matsuura, 2006).

Fluxes determined at Esieh Lake with the MOD chamber were compared to a direct bubble trap measurement in an area of the lake that was approximately 4 m deep. The bubble trap consisted of an octagonal pyramid, with an open base area of 45 $m^2$ and approximate height of 2 m. The walls of the pyramidal were made of plastic sheeting and fixed on a PVC tubular structure (Figures 4 and S3). At each corner of the structure, an anchor and a float were fixed in such a manner that the funnel was steadily positioned under water at about 1 m above sediments. At the center of the funnel, an additional float was

fixed to keep the pyramid taut. At the top of the bubble trap, a specially manufactured union connected the inner volume of the pyramid to a straight 2″ pipe, of 3 m length that was kept in vertical position. A Pitot tube with a high frequency 0–500 Pa differential pressure sensor was used to measure the gas speed, and therefore the gas flow rate collected by the bubble trap. The $CH_4$ flow was formed by discrete bubbles, and as such, the signal had a high degree of both high-frequency and low-frequency variability. Flow data was recorded for 2 second each minute, to filter out high-frequency signals. Multiple

minutes worth of samples were averaged in order to determine the actual flow rate. The bubble trap in operation was calibrated by comparing the measured flow rate and the time required to fill a 155 L plastic bag. This bubble trap was deployed on August 27, 2018; i.e. just after MOD chamber deployment.

In addition to field testing at Esieh Lake, the impact of motion on diffusive flux measurements was quantified using the same chamber in a section of "Lago de Guadalupe" where no ebullition was observed. In this case, the MOD chamber was

operated with a continuous flow of $CH_4$-free nitrogen, exactly as the ODC method (Gerardo-Nieto et al., 2019), from a small guiding boat powered by an electrical fishing engine with speed control. The chamber was kept stationary for several minutes during which $F$ was measured constantly; then, the chamber was put in motion at a speed of approximately 0.56 m s⁻$^1$ (2 km h⁻$^1$), intentionally above the maximum speed reached during the transects at Esieh Lake. The chamber was kept in motion until relatively stable readings were obtained. It is worth noting that this method was applied in "Lago de Guadalupe"

instead of Esieh lake because the ODC method used (with the MOD chamber) required and electric powered boat, compressed gas and gas flow control, unavailable at the remote location of Esieh Lake.





## 3 Results and discussion

During the field campaign, the MOD chamber was first tested in a still position in a region of Esieh Lake where high ebullition was observed. During this first test, keeping the chamber exactly over an ebullition hotspot was identified as a

difficult task, due to boat motion caused by wind and waves, but also because large bubble seeps generated strong radial water movement at the surface, pushing the chamber outward away from the center of hotspot seeps. In addition, even when a gas burst was captured by the chamber, the airflow sensor did not produce a clear signal among the large noise. We linked that noise to stochastic ebullition and strong agitation caused by the seep, causing pressure and flowrates oscillations. In contrast, we observed that, when in motion, the chamber was able to cross hotspots without being diverted, probably thanks

to the keels and the kinetic energy of the chamber in straight motion. Due to the difficulties of maintaining stationary positions and accurately measuring $Q_0$, the measurement of high ebullition at stationary location was quickly abandoned in favor of ebullition flux measurements in motion, which does not require flowrate measurement.

During the field campaign we measured $CH_4$ emissions in a selected 3500 $m^2$ macroseepage area (Movie S1), where strong ebullition was observed. In that section, 15 transects for a total of 72 flux measurements were done. We also made eight

stationary diffusive flux measurements next to the macroseepage area where ebullition was not observed. Paddling under variable wind velocities and directions made it difficult to maintain a constant boat speed along the transects. Overall, during the 15 transects, the mean transect speed, determined from the distance between the length and the duration of the transect; i.e. not relying on imprecise GPS speed indicators, ranged from 0.19 to 0.50 m s$^{-1}$, with a mean of $0.30 \pm 0.09$ (mean $\pm$ one standard deviation of the mean). It should be highlighted that the speed during transect has no effect on the method, except

the effect that motion has on diffusive fluxes, which will be discussed later. The distance covered by each transect was $42 \pm 14$ m. A typical example of the results obtained during a transect is shown in Fig. 5A, during which four sharp $C_D$ increases were detected. A total of 10 of these abrupt $C_D$ increases were used to fit Eq. (S2) and to determine $\theta$, which was relatively constant at $11.35 \pm 3.13$ s (results not shown) with a coefficient of determination ($R^2$) of $0.991 \pm 0.007$.

From the same data set, the instantaneous flux was determined using Eq. (5), and is presented in Fig. 5B. As shown, despite

double data smoothening, a significant noise was still observed. The exact contributions of the lake $CH_4$ flux and the MOD chamber method to that noise is uncertain, although differences in noise were observed within a single transect or among different transects, which indicates that part of the noise was caused by the lake bubbling dynamics. Despite noise, the mean $F$ measured from Eq. (5) during transects was equal to those estimated from Eq. (6) and had therefore no impact on overall $F$ determinations. The mean $F$ determined from the transects in the selected ebullition zone of the lake was highly variable, as

shown on Fig. S4.A, and ranged from $3.4 \times 10^1$ to $2.8 \times 10^4$ g $CH_4$ m$^{-2}$ d$^{-1}$; i.e. over 3 orders of magnitude, with a mean and standard deviation of the mean of $2518 \pm 5379$ g $CH_4$ m$^{-2}$ d$^{-1}$.

To confirm the potential of the MOD chamber to also measure diffusive fluxes we conducted stationary measurements adjacent to the macroseepage site, but in an area where no clear ebullition was observed. On average, the mean diffusive flux from the water adjacent to the macroseepage area was $27.5 \pm 21.6$ g m$^{-2}$ d$^{-1}$ (data not shown). This is about 3 orders of




magnitude above mean diffusive fluxes from lakes north of 66° N (Wik et al., 2013; Bastviken et al., 2011), and suggests that the intense ebullition observed promotes $CH_4$ transfer to the water column and triggers diffusive fluxes.

We created a $CH_4$ emission map by interpolating data collected from these transects and stationary measurements (Fig. 6). Mean emission from the interpolated data was of 1226 g $CH_4$ $m^{-2}$ $d^{-1}$, which corresponds to a total daily emission of 4.2 tons of $CH_4$ over the entire 3500 $m^2$ lake section that was selected. Five hours of continuous measurement of the gas collected by

the bubble trap on August 27 showed a highly variable flowrate of $31 \pm 34$ L $min^{-1}$. The $CH_4$ content of the collected bubble gas was determined by gas chromatography (83.4 % v/v; J. Chanton's laboratory, Florida State University), which allowed to determine a mean $CH_4$ emission of $575 \pm 618$ g $CH_4$ $m^{-2}$ $d^{-1}$ (Fig. S5). In order to better compare emissions determined by the MOD chamber and the bubble trap, the position of the latter was localized on the $CH_4$ emission map with a 4 m error range, which reflects potential coordinates error between MOD chamber and bubble trap deployments, and is represented by

the red discontinuous circle in Fig. 6. When considering any position of the bubble trap within that circle, the mean map-based emission; i.e. determined from interpolated MOD chamber measurements, was $542 \pm 522$ g $CH_4$ $m^{-2}$ $d^{-1}$, thus showing no significant difference between the MOD and bubble-trap methods. The mean emission in the selected section of Esieh Lake is within reported range for seepage (Etiope, 2015), 4 orders of magnitude higher than mean emissions from lakes northward of 66°N (Bastviken et al., 2011; Wik et al., 2013), and 2 orders of magnitude above the mean emission reported

for wetlands (Kayranli et al., 2010). The total daily emission of the selected area, estimated to 4.2 tons of $CH_4$, is of the same magnitude as emissions reported for macroseepage globally (Etiope, 2015).

From the emission determined in the selected section of Esieh Lake, considering 83.4 % $CH_4$ content in bubbles, the gas emission flowrate observed during transect measurements ranged from $5.8 \times 10^1$ to $4.9 \times 10^4$ L $m^{-2}$ $d^{-1}$ or 0.04−34 L $m^{-2}$ $min^{-1}$. Similarly, from the abrupt $C_C$ increases, the $CH_4$ content of the bubbles ($M_B$) and their size ($d_B$) was determined, which is a

potential additional benefit of the MOD chamber. These parameters are indeed the most important parameters that affect ebullitive $CH_4$ transport through the water column and to the atmosphere (DelSontro et al., 2015; Greene et al., 2014). Overall, $M_B$ ranged from 1.23 to 781 mg $CH_4$ with a mean of $81 \pm 144$ mg $CH_4$. The corresponding bubble diameter ranged 16−138 mm, which is within standard bubble diameter considered in seep flux estimations (Etiope et al., 2004) but does not match visual observations since numerous small bubbles were observed at Esieh lake. This suggest that the MOD chamber

method does not detect and quantify small bubbles, although these small bubbles are included in $F$ measurements. Small bubbles might be the reason why high flux noise was observed during transect measurements (Fig. 5B).

The ebullitive flowrate reaching the MOD chamber ($Q_0$) was determined to $0.33 \pm 0.71$ L $min^{-1}$, thus validating the condition $Q_0 < Q_D$ that was considered during this work. However, it is worth noting that, in five occasions over a total of 74 flux measurements, $Q_0$ was temporarily exceeding $Q_D$, showing that the MOD chamber, as applied in Esieh Lake, was reaching

its overall maximum flux measurement capacity. Considering the condition $Q_0 = Q_D$ as the frontier condition, the prototype configuration used allowed for the measurement of a maximum steady flux of $8.4 \times 10^3$ to $1.08 \times 10^4$ g $CH_4$ $m^{-2}$ $d^{-1}$ (equivalent to 10.1−13.0 L $m^{-2}$ $min^{-1}$ of 83.4 % $CH_4$ bubbles), with the UGGA and the EX-TEC detector, respectively. Nonetheless, it should be kept in mind that this upper limit of the MOD design is proper to the chamber design tested at





Esieh Lake. Indeed, an additional extraction pump, working at a precise flowrate, could be added to the configuration of the
MOD chamber, without any modification of the mass balance equations (if $C_C$ is kept far above $C_{ATM}$). That additional pump
would allow increasing $Q_D$ above the flowrate extracted by the detector and make the MOD chamber applicable to
practically unlimited flux intensity. However, it is of crucial importance for $Q_D$ to be well known, to avoid error in the mass
balance. Thus, the additional pump used should ensure constant flowrate and be precisely calibrated. From our results, these
arguments, and the previously established difficulty to measure macroseepage hotspots at fixed locations, we conclude that
the MOD chamber working at $Q_0 < Q_D$, is a resourceful option for seeps measurements.

An important point left to discuss is the effect of chamber motion on the diffusive component of $CH_4$ flux. Any water
movement affects the gas/liquid boundary layer, which is of crucial importance in mass transfer (Schubert et al., 2012; Lorke
et al., 2015). By using an expressly designed chamber, we tried to reduce this impact, but still, the diffusive $CH_4$ flux during
rowing transects might have been overestimated at Esieh Lake. To test that impact, the MOD chamber was deployed in a
region of Lago de Guadalupe where no ebullition was observed. We compared the flux measured continuously in a
stationary, drifting position without rowing and then in motion at approximately 0.56 m s$^{-1}$, which is on purpose above the
maximum speed during transects in Esieh Lake. The results, presented in Fig. S6, show measured fluxes in a relative scale.
We observed that $F$ during motion was 2.1 to 3.4 times higher compared to $F$ under stationary conditions and the mean
increase factor was 2.8, confirming that the chamber motion decreases the boundary layer thickness and artificially increases
the diffusive flux. It should be noted, that during these tests, the speed indicator used was our GPS, which is highly unprecise
at such a low speed, and might explain at least part of the important noise observed during motion, as well as differences
between replicates.

To quantify that impact on $F$ measured at Esieh Lake, we isolated diffusive fluxes events, observed during transects. Indeed,
transects and $F$ determinations were started in regions of the lake where little or no ebullition was observed, steering toward
regions with high ebullition. Thus, in many cases, the initial measurements were done while moving but with diffusive flux
only. These diffusive fluxes ranged 4.8−230 g m$^{-2}$ d$^{-1}$, with a mean of 64 ± 57 g m$^{-2}$ d$^{-1}$, ($n$ = 14; Fig. S4.B). This mean
diffusive flux represented 2.56 % of the mean total flux measured during transects, thus, if this diffusive flux was
overestimated 2.8 times (as shown on Fig. S6), the error committed would have been an overestimation of 1.65 % of the total
$F$.

The new concept of a dynamic chamber moving at the surface of a lake showed several benefits for the measurement of
emissions from lakes with intense $CH_4$ ebullition seeps. The main features that makes the MOD chamber of interest is that
while moving, each point along a transect is sampled with the same statistical relative weight. It dispenses with the complex
chamber positioning over hotspots, does not require the measurement of the gas flowrate emitted by the lake, and does not
involve an arbitrary classification of individual seeps. Unlike the large bubble trap which was limited to a fixed position and
required four people for fieldwork, the MOD was lightweight and easily operated by one person. We demonstrated
experimentally that this method allows for the measurement of emissions of up to 1.08 × 10$^4$ g $CH_4$ m$^{-2}$ d$^{-1}$. However, this
theoretical border is not a fixed limit, since the addition of an extractor with a higher flowrate would allow, in theory,





measurement of higher emission magnitude. We also confirmed that the same chamber could be used for low diffusive fluxes, which is not surprising as the MOD chamber is similar to static chambers when operated as a closed loop under static

position. Thus, the MOD chamber is versatile by covering the entire magnitude range of $CH_4$ emissions currently identified in aquatic systems. A comparison to other methods (i.e. echo-sounders) is difficult, because to the best of our knowledge, none of them have been used to measure macroseeps such as those found at Esieh Lake. However, among those that could be theoretically used with the same purpose (Table S1), the MOD chamber is the single method allowing measurements under motion or static position and covering the entire range of aquatic ecosystem emissions, from low diffusive flux to large

ebullition seeps, with a single apparatus. Regarding costs, the MOD chamber is moderately expensive. It requires a $CH_4$ analyzer, which is the costliest component, ranging from about 10,000$ (cost of the Ex-Tec) to 50,000 US$ (cost of the UGGA). However, any $CH_4$ detector with moderate sensitivity could serve, and a low-cost detector as the model used by Duc et al. (2020) could be convenient.

Despite the promising results obtained, we acknowledge that the concept tested would greatly benefit from further research.

First, we observed that chamber motion affects diffusive flux measurements, at least to a minor extent, in such manner that the development of a precise speed controlling device as well as a more systematic evaluation of speed impact would be of great interest. Second, the MOD chamber design was conceptually developed from a trial/error approach, which included testing several chamber designs and shapes. A more systematic direct engineering approach, for instance, including computational fluid dynamics studies, might lead to an improved design. Hence, the design suggested here is already

operational and field validated for its use on lakes with $CH_4$ ebullition seeps during ice free conditions. Its potential use under other configurations might be foreseen. For example, the use of the same concept on bubble-induced open holes in lake ice might be considered. In this case, the chamber should be used in stationary position, which has been shown difficult in the present work, but the presence of ice around the seeps might greatly facilitate firm positioning, thus avoiding the problem encountered at Esieh Lake

**4 Conclusion**

The method suggested here is operational and field validated for its use on lakes with $CH_4$ emissions ranging the entire magnitude of $CH_4$ emissions currently identified, from those standardly observed in lakes to intense macroseeps, with a single apparatus of moderate cost. The MOD chamber is a promising method for the determination of seepage in aquatic environments, not only with the objective to update current seepage estimations, but also to monitor their expected increase

as permafrost thaws and large gas seeps potentially become more abundant in the future.

*Data availability*. The data associated with this work are stored at; Thalasso, Frederic (2020), "MOD chamber HESS data", Mendeley Data, V1, doi: 10.17632/kpbc6mhwjt.1



*Supplement*. The Supporting Information includes 6 pages, 6 Figures, 1 movie, 1 Table and 6 equations.

Author contribution. FT and KWA conceived the study. FT wrote the paper. FT and RGV analyzed the MOD data. OI, EC, and LB were responsible for the large bubble trap analysis. All authors except RGV contributed to field work. All authors have given approval to the final version of the manuscript.


*Competing interests*. The authors declare that they have no conflict of interest.

*Acknowledgments.* The authors thank Francisco Silva-Olmedo, David Flores-Rojas, and Andrés Rodríguez Castellanos for their technical assistance. Janelle Sharp, Oscar Gerardo-Nieto, and Yameli Alfano-Ojeda assisted with field work. The
NANA Regional Corporation granted permission and logistical support for field work at Esieh Lake. This work was supported by the "Consejo Nacional de Ciencia y Tecnología" (Conacyt, project 255704) and NASA ABoVE NNN12AA01C.

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




**Figure captions**

**Figure 1: Conceptual sketch of the mobile open dynamic (MOD) chamber, shown at the surface of the lake, passing over an intense seep (A); Dimensions (cm) of the prototype built and used in the present work (B). Darker and lighter blue colors indicate three aluminum sheets welded together.**


**Figure 2: Mass balance of the MOD chamber (see text for details).**

**Figure 3: Prototype being operated on Esieh Lake (Credit: K. Walter Anthony).**

**Figure 4: Bubble trap shown during installation before it was submerged (A) and during operation (B); the inflatable boat contained the measurement device at the center of the submerged bubble trap (Credit: O. Irzak).**

**Figure 5: Typical example of; (A) $C_D$ (grey solid line) and $C_C$ (black solid line) measured during a transect, and (B) instantaneous flux computed from these concentrations. Blue arrows show when large bubbles were captured by the chamber and red marker**
**shows an example of $\Delta C_C$ used to determine the $CH_4$ content of the bubbles (see Section S1). Please note the logarithmic scales.**

**Figure 6: Map of $CH_4$ emissions in a region of Esieh Lake with large gas seeps; black crosses (+) indicate central location of transects measurements. This map scale is metric; i.e. same distance scale in both axes. Please note the logarithmic color scale. Red octagon indicates the location of the bubble trap, while the dotted red circle represents potential coordinates error between MOD**
**chamber and bubble trap deployments; i.e. ± 4 m, see text for details).**



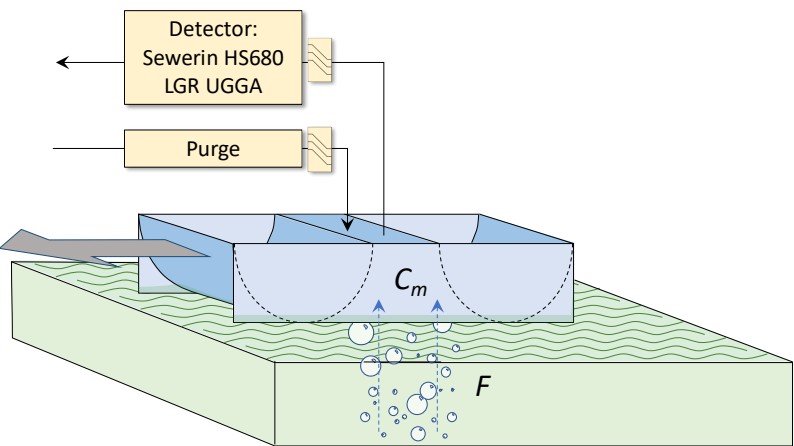


**Figure 1: Conceptual sketch of the mobile open dynamic (MOD) chamber, shown at the surface of the lake, passing over an intense seep (A); Dimensions (cm) of the prototype built and used in the present work (B). Darker and lighter blue colors indicate three aluminum sheets welded together.**





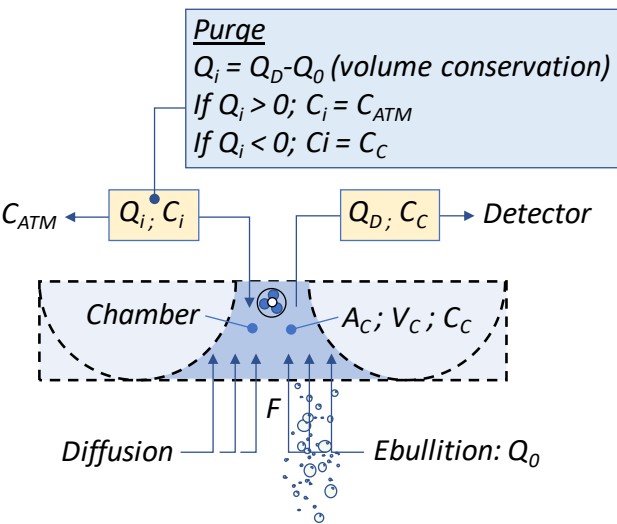

**Figure 2: Mass balance of the MOD chamber (see text for details).**




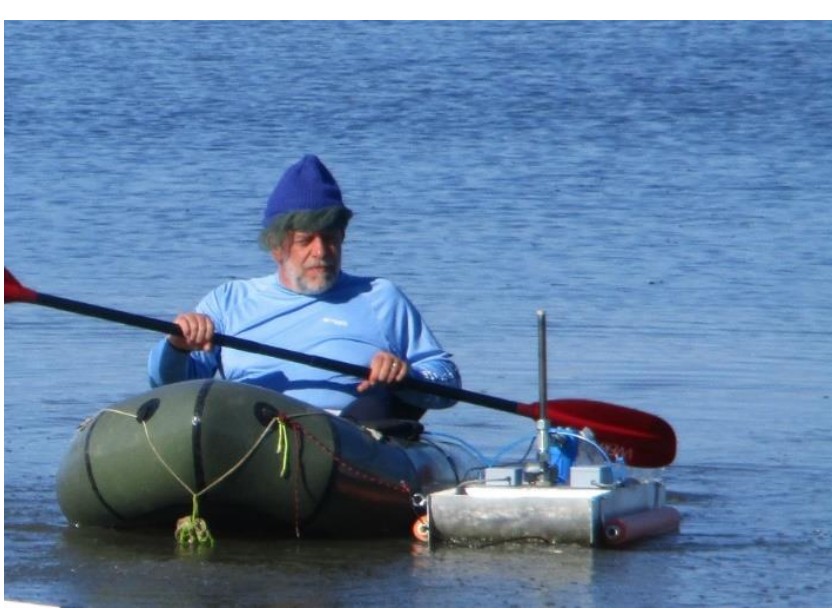

**Figure 3: Prototype being operated on Esieh Lake (Credit: K. Walter Anthony).**





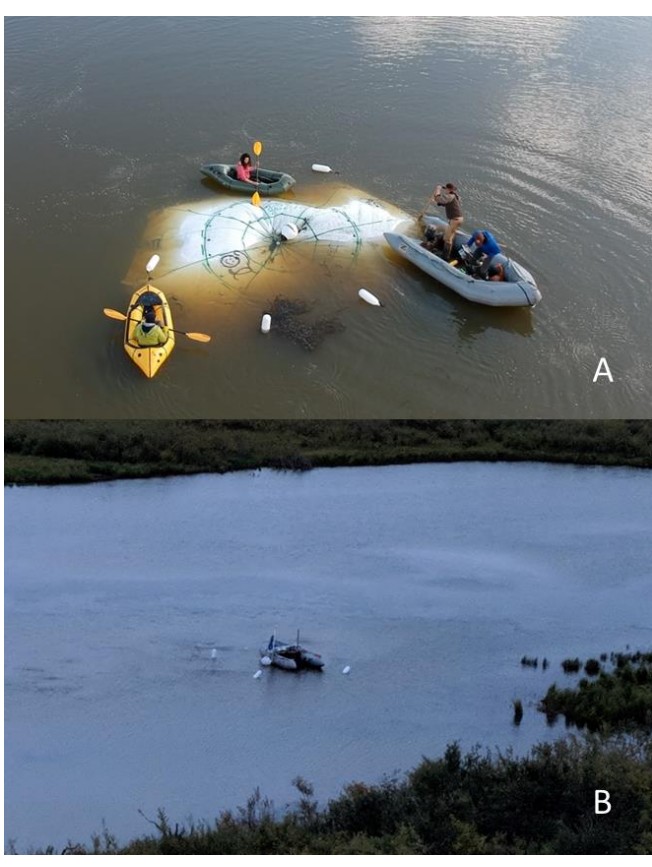


**Figure 4: Bubble trap shown during installation before it was submerged (A) and during operation (B); the inflatable boat contained the measurement device at the center of the submerged bubble trap (Credit: O. Irzak).**






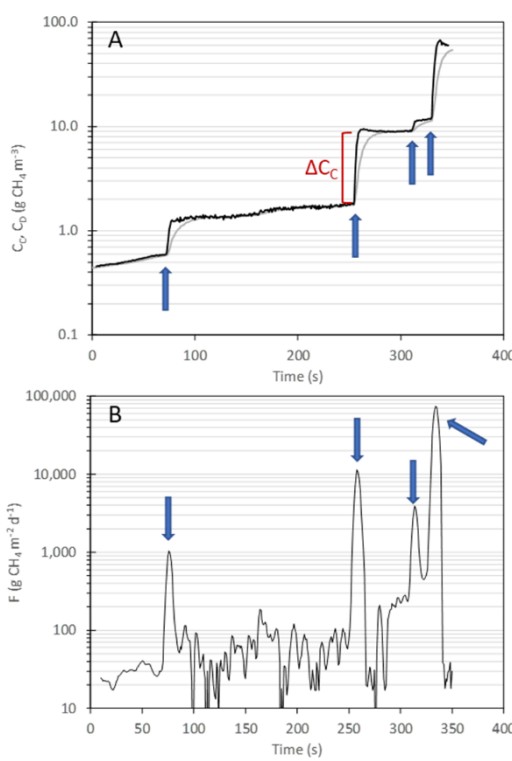

**Figure 5: Typical example of; (A) $C_D$ (grey solid line) and $C_C$ (black solid line) measured during a transect, and (B) instantaneous flux computed from these concentrations. Blue arrows show when large bubbles were captured by the chamber and red marker shows an example of $\Delta C_C$ used to determine the CH$_4$ content of the bubbles (see Section S1). Please note the logarithmic scales.**




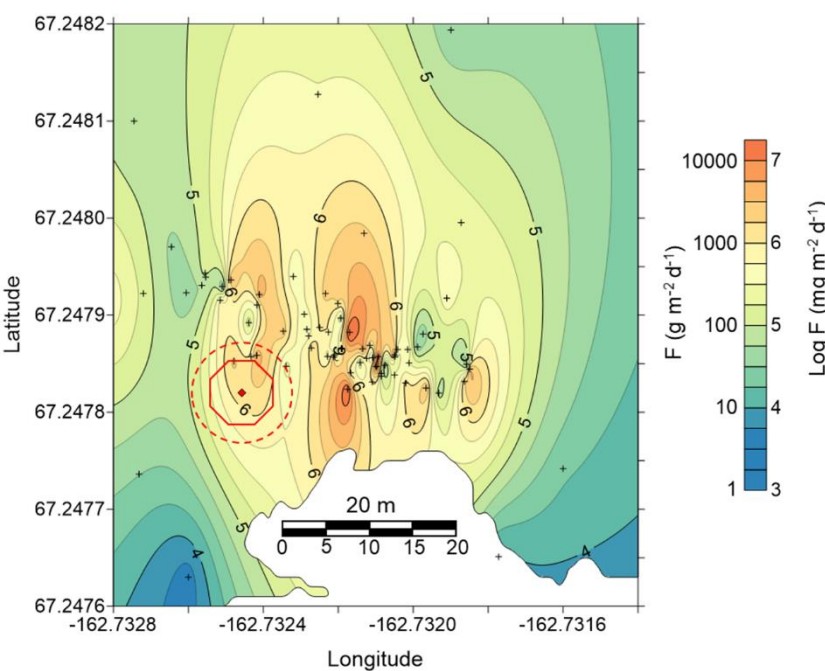

**Figure 6: Map of CH₄ emissions in a region of Esieh Lake with large gas seeps; black crosses (+) indicate central location of transects measurements. This map scale is metric; i.e. same distance scale in both axes. Please note the logarithmic color scale. Red octagon indicates the location of the bubble trap, while the dotted red circle represents potential coordinates error between MOD chamber and bubble trap deployments; i.e. ± 4 m, see text for details).**