# Peer review of "Mobile open dynamic chamber measurement of methane macroseeps in lakes"

_Hydrology and Earth System Sciences, 2020_

## Referee Comment (RC1) · Anonymous Referee #1 · 13 Oct 2020

This ms covers at least four important research issues on greenhouse gases (GHG) in freshwater systems:

1. Seepages in freshwater lake's sediments 2. Estimation of emissions in systems with unevenly distributed hotspots 3. Development of an easy-to-use instrument with high precision in a wide concentration range 4. Measurements in remote areas

The development and successful use of a newly developed flux chamber is in the center of the paper by F. Thalasso et al., the so-called Mobile open dynamic chamber (MOD). A permanent flow of gas makes the chamber to an open system, the gas of interest (CH4, CO2) is emitted across the air-water-interface (AWI) into a cavity, from where it is transported by the continuous flow of ambient air to the GHG-analyser (no matter of which brand). The chamber can be connected to a boat manned or unmanned

and moved along a system of transects across the interesting area of an aquatic system, e.g. a lake with seeps. The authors represent a set of equations to calculate the emission rates for systems based on data generated by MOD, as well as assembly instructions for the chamber. The ms will allow interested scientists to build and use a MOD and further on, to calculate gas emissions. This easy-to build design and the abandonment of special gases in tanks allow usage in remote places, like the tested Alaskan lake. For test and calibration purposes, a second lake in Mexico as well as a very specific gas trap were involved. While L. Guadalupe represents another system with low or medium ebullition rates, the tested trap is unique, already by its trapping area of 45 m$^2$. It is interesting that the results fit between MOD and "giant trap", for the idea of using simple repeatable techniques with low man power in a remote area, a few simple inverted funnels would have been more useful. As mentioned by the authors, new small and reasonable sensors (CH4 and CO2) become more and more available and could replace expensive gas-analysers soon. I can imagine that in near future slightly modified MODs equipped with sensors and data-storages and connected to remote vehicles will help to solve the problem of spatial and temporal resolution of emission rates from freshwater lakes. Following equations S4-6 (supporting information) the authors calculate the spherical diameter of gas bubbles rising to the AWI, based on the methane content in bubbles. This content has to be analysed in parallel (here by gas chromatography). From my point of view these calculation simplifies the bubble size calculation and additionally, as it is known that GHG content within bubbles can vary by several per cent, another unknown is in the equation. Perhaps I oversee it, but I am missing water depth/pressure at depth in these equations as well as size when leaving the sediment. I recommend to rewrite this paragraph to clarify the calculation pathway. The ms is well written without typos. In conclusion, all four approaches mentioned above, are treated in this manuscript and contribute to these fields in GHG research.

420, 2020.

---

## Referee Comment (RC2) · Anonymous Referee #2 · 19 Oct 2020

Comments to hess 2020-420

The manuscript introduces a mobile open dynamic chamber to determine the methane flux from lakes, ebullitive or diffusive. The Ms is well written and is clear and concise. However, there are some aspect which I found difficult to follow:

The design of the MOD is a bit difficult to grasp. More photos could be helpful here. Figure 1 should be improved, why is the top of the chamber open? What sort of purge ventil is used?

I found the indices for the different parameters a bit confusing. D- for detector and C- for chamber, ok, but I and O ??

I was wondering, when the detector is sucking gas from the chamber and there is no

ebullition, the purge ventil would let air into the chamber. I do not understand how you correct for this dilution and how you realize when there is more ebullition or more dilution? Ok, I read now the details on the purge ventil. But still, did you correct your measured data for a possible dilution?

Why did you divide your measurements into segments? The nice thing about your MOD is that you have continuous measurements on the whole lake....

Some more examples for the measured data and calculated fluxes would be appreciated.

---

## Author Response (AR1)

* * *
This ms covers at least four important research issues on greenhouse gases (GHG) in freshwater systems:

1. Seepages in freshwater lake's sediments 2. Estimation of emissions in systems with unevenly distributed hotspots 3. Development of an easy-to-use instrument with high precision in a wide concentration range 4. Measurements in remote areas

The development and successful use of a newly developed flux chamber is in the center of the paper by F. Thalasso et al., the so-called Mobile open dynamic chamber (MOD). A permanent flow of gas makes the chamber to an open system, the gas of interest ($CH_4$, $CO_2$) is emitted across the air-water-interface (AWI) into a cavity, from where it is transported by the continuous flow of ambient air to the GHG-analyser (no matter of which brand). The chamber can be connected to a boat manned or unmanned and moved along a system of transects across the interesting area of an aquatic system, e.g. a lake with seeps. The authors represent a set of equations to calculate the emission rates for systems based on data generated by MOD, as well as assembly instructions for the chamber. The ms will allow interested scientists to build and use a MOD and further on, to calculate gas emissions. This easy-to build design and the abandonment of special gases in tanks allow usage in remote places, like the tested Alaskan lake. For test and calibration purposes, a second lake in Mexico as well as a very specific gas trap were involved. While L. Guadalupe represents another system with low or medium ebullition rates, the tested trap is unique, already by its trapping area of 45 m2. It is interesting that the results fit between MOD and "giant trap", for the idea of using simple repeatable techniques with low man power in a remote area, a few simple inverted funnels would have been more useful. As mentioned by the authors, new small and reasonable sensors ($CH_4$ and $CO_2$) become more and more available and could replace expensive gas-analysers soon. I can imagine that in near future slightly modified MODs equipped with sensors and data-storages and connected to remote vehicles will help to solve the problem of spatial and temporal resolution of emission rates from freshwater lakes.

Our answer: Thank you for the positive appreciation of our work. We are pleased that Anonymous Referee #1 found our manuscript of interest.

Technical comment: "Following equations S4-6 (supporting information) the authors calculate the spherical diameter of gas bubbles rising to the AWI, based on the methane content in bubbles. This content has to be analysed in parallel (here by gas chromatography). From my point of view these calculation simplifies the bubble size calculation and additionally, as it is known that GHG content within bubbles can vary by several per cent, another unknown is in the equation. Perhaps I oversee it,

but I am missing water depth/pressure at depth in these equations as well as size when leaving the sediment. I recommend to rewrite this paragraph to clarify the calculation pathway".

Our answer: We agree that pressure should be specified in our calculations. We have clarified this section of the supporting information as follows;

"From $M_B$, the volume of the bubbles ($V_B$) and their equivalent spherical diameter ($d_B$) at atmospheric pressure were determined, assuming that the $CH_4$ content in the bubbles ($\%_{CH4}$) is known, according to Eq. (S5) and (S6), respectively.

$$V_B = \frac{M_B}{16} \cdot \frac{R \cdot T}{P} \cdot \frac{1}{\%_{CH4}} \tag{S5}$$

$$d_B = 2 \cdot \sqrt[3]{\frac{3 \cdot V_B}{4 \cdot \pi}} \tag{S6}$$

where 16 is the molecular weight of $CH_4$ (g), $R$ is the universal gas constant (L atm mol$^{-1}$ K$^{-1}$), $T$ is the temperature (K) and $P$ is the atmospheric pressure (atm).

Since bubble volume and diameters are important for mass transfer determination during their migration to the lake surface, the actual bubble volume ($V'_B$) at a given depth ($D$) within the water column is given by Eq. (S7).

$$V'_B = V_B \cdot \frac{P}{\frac{(\rho \cdot g \cdot D)}{101,325} + P} \tag{S7}$$

where $\rho$ is the water volumetric mass density (kg m$^{-3}$), g is the standard gravity (m s$^{-2}$), and 101,325 is the conversion factor from Pa to atm.
The manuscript introduces a mobile open dynamic chamber to determine the methane flux from lakes, ebullitive or diffusive. The Ms is well written and is clear and concise. However, there are some aspect which I found difficult to follow:

Our answer: Thank you for this kind appreciation of our work.

The design of the MOD is a bit difficult to grasp. More photos could be helpful here. Figure 1 should be improved, why is the top of the chamber open? What sort of purge ventil is used?

Our answer: We appreciate this comment. It is important to ensure the reader clearly understands the MOD chamber design. We attended this comment by including an additional cross-section in Figure 1, which indicates that the chamber is closed in its upper part. We also included two more pictures of the chamber in the supporting information, as new Figure S3. Please note that the purge design is described around L190.

I found the indices for the different parameters a bit confusing. D- for detector and C for chamber, ok, but I and O ??

Our answer: Thank you for this comment, we agree. We changed subscript i for P (standing for purge) and subscript 0 to B (standing for bubbles/ebullition). We also included a new Table of notation in supporting information. We hope this is clearer now.

I was wondering, when the detector is sucking gas from the chamber and there is no ebullition, the purge ventil would let air into the chamber. I do not understand how you correct for this dilution and how you realize when there is more ebullition or more dilution? Ok, I read now the details on the purge ventil. But still, did you correct your measured data for a possible dilution?

Our answer: Yes indeed, our mass balance equations consider the flow rate that is entering the chamber when ebullition is low (Eq. 4). However, we also indicate that the input of $CH_4$ through the purge is negligible, compared to the CH4 emitted by the lake (Eq. 5 and text related).

Why did you divide your measurements into segments? The nice thing about your MOD is that you have continuous measurements on the whole lake…

Our answer: This is correct, we divided our transects into segments, and for each of them we determined a mean emission. The main reason for this approach is the stochastic nature of ebullitive events, generating instantaneous flux varying several orders of magnitude in matter of seconds (Figure 5 and new Figure S5). The interpolation of instantaneous fluxes would be extremely difficult to perform and, in our opinion, of moderate interest as being a representation of a very specific instant. On the contrary, the mean emission measured during each segment (typically 10 m long), gives the mean emission in the corresponding region of the lake that can be easily interpolated.

Some more examples for the measured data and calculated fluxes would be appreciated.

Our answer: We agree. We have included a second example of measurement observed during a transect, presented in Supporting information as new Figure S5.

New text included (in red):

Figure 1: Conceptual sketch of the mobile open dynamic (MOD) chamber, shown at the surface of the lake, passing over an intense seep (A); Cross section of the chamber cavity (B). Darker and lighter blue colors indicate three aluminum sheets welded together.

Figure S3. Superior (A) and inferior (B) view of the chamber hull with lateral floats added for improved stability.

[revised manuscript text omitted]

Number of Tables: 2
Equations from S1 to S7

**Table S1**: Notations.

| Notation | Description | Units |
|---|---|---|
| A | Area | $m^2$ |
| C | Gas concentration | $g\ m^{-3}$ |
| d | Diameter | m |
| F | Gas flux | $g\ m^{-2}\ s^{-1}$ |
| g | Standard gravity | $m\ s^{-2}$ |
| M | Mass | g |
| P | Pressure | atm |
| Q | Gas flowrate | $m^3\ s^{-1}$ |
| R | Universal gas constant | $L\ atm\ mol^{-1}\ K^{-1}$ |
| T | Temperature | K |
| V | Gas volume | $m^3$ |
| $\%_{CH4}$ | Volume percentage of $CH_4$ in bubbles | % |
| $\theta$ | Response time | s |
| $\rho$ | Volumetric density | $kg\ m^{-3}$ |
| **Subscripts** | | |
| P | Purge | |
| B | Bubble or ebullition | |
| D | Detector | |
| C | Chamber | |
| ATM | Atmospheric | |
| t | Time | |

[Figure]

**Figure S1**. Methane bubbles trapped in the ice of an arctic lake, illustrating that ebullition occurs repeatedly in specific locations (Credit: A. Sepulveda-Jauregui, F. Thalasso).

[Figure]

**Figure S2**. Dimension of the prototype built and used in the present work. Darker and lighter blue colors indicate three independent aluminum foils welded together. Dimensions are in cm.

[Figure]

**Figure S3**. Superior (A) and inferior (B) view of the chamber hull with lateral floats added for improved stability.

[Figure]

**Movie S1**. Methane seeps; general and closeup views. Available at: Thalasso, Frederic (2020), "Esieh lake seepage HESS", Mendeley Data, V1, doi: 10.17632/fnr3mkxmk9.1

**S1. Response time and data interpretation**

The concentration read by the detector has a certain delay, due to the gas residence time from the chamber to the detector. However, if the detector is close to the chamber and the tubing of a reduced diameter, this time is very short; i.e., from 1.6 to 2.0 s in our case. However, even if it can be assumed that a bubble entering the chamber is immediately mixed within the chamber, the detectors have an inherent response time. This effect causes a certain delay and a buffer time, between the actual concentration read by the detector ($C_D$) and $C_C$. To take this delay into account a standard mixing model can be used (Eq. S12), where $\theta$ is the response time of the system

$$C_C = \left(\frac{dC_D}{dt} \cdot \theta\right) + C_D \tag{S1}$$

In Eq. (S1), $\theta$ was determined from experimental data, using several step $C_D$ increases observed in the field. The adjustment was done through excel, minimizing the Root Mean Square Error (RMSE) between experimental $C_D$ data and Eq. (S2), where $C_{D,0}$ is the initial reading of the detector (at time 0), and $C_C$ is the actual concentration in the chamber.

$$C_D = C_{D,0} + \left[(1 - \exp(-t/\theta)) * (C_C - C_{D,0})\right] \tag{S2}$$

After $C_C$ was determined, Eq. (5) was used to determine instantaneous $F$ along the transects. Alternatively, Eq. (6) was used to determined mean flux over a transect section. In the case of instantaneous F, during transects, and despite the relatively high signal to noise ratio of detectors used; i.e., ratio of the mean to the standard deviation, $F$ was subject to a significant noise, and a first data smoothening of $C_C$ was necessary, followed by a second smoothening of $dC_C/dt$ (Eq. S7). In both cases we opted for a pondered smoothening described by Eq. S3, where X′ is the smoothened variable X, in this case $C_C$ or $dC_C/dt$.

$$X'_t = 0.1 \cdot X_{t-2} + 0.2 \cdot X_{t-1} + 0.4 \cdot X_t + 0.2 \cdot X_{t+1} + 0.1 \cdot X_{t+2} \tag{S3}$$

As it will be shown in the Results and Discussion section, peak fluxes were detected, which corresponded to step increases of $C_C$ ($\Delta C_C$), caused by bubbles reaching the chamber. These abrupt increases offer a unique opportunity to quantify the $CH_4$ mass content of the bubbles ($M_B$). It should be noticed that since these step increases were observed in a few seconds, the amount of $CH_4$ lost through detector extraction or entering the chamber can be neglected over that short time, as far a as single and clear increase was observed. Thus, $M_B$ was determined during the field experiment according to Eq. (S4).

$$M_B = \Delta C_C \cdot V_C \tag{S4}$$

From $M_B$, the volume of the bubbles ($V_B$) and their equivalent spherical diameter ($d_B$) at atmospheric pressure were determined, assuming that the $CH_4$ content in the bubbles ($\%_{CH4}$) is known, according to Eq. (S5) and (S6), respectively.

$$V_B = \frac{M_B}{16} \cdot \frac{R \cdot T}{P} \cdot \frac{1}{\%_{CH4}} \tag{S5}$$

$$d_B = 2 \cdot \sqrt[3]{\frac{3 \cdot V_B}{4 \cdot \pi}} \qquad\qquad (S6)$$

where 16 is the molecular weight of $CH_4$ (g), $R$ is the universal gas constant (L atm mol$^{-1}$ K$^{-1}$), $T$ is the temperature (K) and $P$ is the atmospheric pressure (atm).

Since bubble volume and diameters are important for mass transfer determination during their migration to the lake surface, the actual bubble volume ($V'_B$) at a given depth ($D$) within the water column is given by Eq. (S7).

$$V'_B = V_B \cdot \frac{P}{\frac{(\rho \cdot g \cdot D)}{101,325} + P} \qquad\qquad (S7)$$

where $\rho$ is the water volumetric mass density (kg m$^{-3}$), g is the standard gravity (m s$^{-2}$), and 101,325 is the conversion factor from Pa to atm.

[Figure]

**Figure S4**. Conceptual sketch of the bubble trap used at Esieh Lake; (A) top view, (B) front view.

[Figure]

**Figure S5**. Additional example of; (A) $C_D$ (grey solid line) and $C_C$ (black solid line) measured during a transect, and (B) instantaneous flux computed from these concentrations. Blue arrows show when large bubbles were captured by the chamber. Please note the logarithmic scales.

[Figure]

**Figure S6**. Box and whiskers showing statistical distribution of fluxes measured with the MOD chamber (A, $n = 74$) and the diffusive component of these fluxes (B, $n = 14$; see text for details). Boxes show interquartile range and median, whiskers represent minimum and maximum, open circles show raw data and filled diamonds represent arithmetic mean.

[Figure]

**Figure S7.** Flux measured by the bubble trap. Each discrete value is the average of 1 minute of continuous measurement. Horizontal discontinuous line shows the mean flux while red continuous line shows 10 minutes moving average of *F* data.

[Figure]

**Figure S8**: Relative fluxes observed with the MOD chamber, under stationary position (left of the arrows) and under motion. Data are presented in relative units, one being the flux observed in stationary position. Horizontal dot-dashed lines represent the mean fluxes during motion.

**Table S2**: Comparison of four methods with a potential to be used in lake seepage.

| | Bubble trap | Duc et al. (2020) | Hydroacoustic | MOD Chamber |
|---|---|---|---|---|
| Large seeps | Yes | Potentially Yes | Potentially Yes | Yes |
| Diffusive flux | No | Yes | No | Yes |
| Mobility | No | No | Yes | Yes |
| Autonomous | No | Yes | No | No |
| Field effort | Important | Moderate | Low | Low |
| Data processing effort | Low | Moderate | High | Moderate |
| Cost range (US$) | Low-cost | Low-cost (un.) | 50000[1] | 10000-50000[2] |

[1]Cost excluding video camera and mounting hardware; [2]Cost of the detector (the cost of the chamber assembly was about 300 US$ in material). un.: undisclosed.